# Effect of Thermoresponsive Xyloglucan on the Bread-Making Properties and Preservation of Gluten-Free Rice-Flour Bread

**DOI:** 10.3390/foods12142761

**Published:** 2023-07-20

**Authors:** Keiko Fujii, Momomi Usui, Akiko Ohsuga, Michiko Tsuji

**Affiliations:** 1Department of Food Science and Nutrition, Faculty of Human Sciences and Design, Japan Women’s University, 2-8-1 Mejirodai, Bunkyo-ku, Tokyo 112-8681, Japanaosuga@fc.jwu.ac.jp (A.O.); 2Department of Health and Nutrition, Faculty of Health and Science, Nagoya Women’s University, 3-40 Shioji-cho, Mizuho-ku, Nagoya-shi 467-8610, Japan; mtsuji@nagoya-wu.ac.jp

**Keywords:** thermoresponsive xyloglucan, tamarind gum, xanthan gum, gluten-free, rice-flour, bread, mechanical properties, sensory evaluation, preservation

## Abstract

This study clarified the effect of adding thermoresponsive xyloglucan on the bread-making properties and preservation of gluten-free rice-flour bread. The thickening polysaccharides used for preparing gluten-free rice-flour bread were modified tamarind gum (MTG; thermoresponsive xyloglucan), tamarind gum (TG), and xanthan gum (XT). The mechanical properties of the added polysaccharide thickener solutions and bread dough, the mechanical properties and sensory characteristics of rice-flour bread, and the aging properties of rice-flour bread were measured. The results showed that the MTG solution exhibited solification at 40 °C and gelation below 40 °C, which affected the dynamic viscoelasticity of the dough. The addition of MTG to gluten-free rice-flour bread reduced the specific volume, increased the moisture content, and reduced the stress at 70% compression. Therefore, the bread with MTG added was soft, moist, and preferred over other those with other additives. In terms of preservation, the addition of 0.5–0.75% of polysaccharides inhibited the hardening and aging of beard with MTG added. This indicates that the addition of MTG at low concentrations is effective in preserving gluten-free rice-flour breads. We found that the thickening polysaccharides had to be added in appropriate concentrations to improve the bread-making properties and achieve the preferred effect.

## 1. Introduction

The food self-sufficiency ratio in Japan has been declining, from 53% in 1985 to 37% in 2020 on a calorie basis [1], which is extremely low even when considering global standards [2]. The annual per capita consumption of rice was 70.0 kg in the fiscal year (FY) 1990 and decreased to 50.7 kg in FY 2020 [3]. Reasons for the decline in rice consumption include the change from a Japanese-style diet of rice and fish to a Western-style diet with increased intake of meat [4] and fat [5], the increasing applicability of bread and noodles [6] as staple foods, and their increasing demand. With this information, the Ministry of Agriculture, Forestry, and Fisheries and many other organizations, including producers’ associations, are working to increase the consumption of domestically produced rice to improve food self-sufficiency. Hence, rice flour made from finely milled rice and rice-flour-based products, such as gluten-free rice-flour bread, have been developed. Moreover, new businesses that utilize such products have been established, increasing the awareness of rice-flour bread [7,8,9,10,11]. However, issues related to taste, price, preservation, and quality have hindered the widespread production of gluten-free rice-flour bread.

Additionally, people suffering from food allergies in Japan account for 1–2% of the total population [12]. Wheat is the second most common food allergen after eggs and milk [13]. There is also a growing international demand for producing gluten-free products for people with celiac disease, a disorder triggered by an immune response to gluten. As a result, wheat-, egg-, and dairy-free products prepared with domestically produced rice flour are attracting attention.

To date, changing the viscosity of the batter by adding thickening agents such as polysaccharides [14,15] or cellulose derivatives has been effective in improving the bread-making properties of gluten-free batter [16,17,18]. However, as a measure to inhibit starch retrogradation, gluten is added to many breads, such as replacement bread [19], by replacing a portion of rice flour with wheat flour (active gluten) [9,20]. Measures for controlling aging are similar, as they involve adding gluten; however, the resulting foods are inappropriate for consumption by people with wheat allergies [21,22]. It was reported that silk fibroin could substitute gluten in preparing 100% rice-flour sponge cake [23]. Additionally, gluten-free rice-flour bread could be prepared using ten types of rice flour by optimizing the amount of water added based on the protein content of the rice flour [24].

Polysaccharide thickeners can impart viscosity, as well as cause gelatinization, enhance the emulsion stability, and enhance other food structure-forming features and the texture. Tamarind seed gum, a polysaccharide thickener, has been shown to enable thickening stabilization, gelation, and ice crystal stabilization and impart starch anti-retrogradation functions to various foods. Hence, tamarind seed gum has been studied for its bread-making properties [25,26] and optimal bread preparation conditions [27]. Recently, a modified tamarind gum (thermoresponsive xyloglucan) with a modified molecular structure of tamarind seed gum was developed. It exhibited characteristics opposite to those of conventional polysaccharide thickeners and could be gelatinized by heating and solubilized by cooling [28,29]. A modified tamarind gum (thermoresponsive xyloglucan) has been widely reported in the fields of tissue engineering and biomedicine, including novel alternatives for tissue regeneration [28,30] and temperature-controlled desorption of cells [31]. Until now, there have been several reports on quality improvement using polysaccharide thickeners in gluten-free rice-flour bread. However, there are only a few specific reports on a modified tamarind gum (thermoresponsive xyloglucan). Therefore, this study aimed to examine the effects of a modified tamarind gum (thermoresponsive xyloglucan) on the bread-making properties and preservation of gluten-free rice-flour bread, as well as the impact on the sensory qualities of the alternative bread.

## 2. Materials and Methods

### 2.1. Materials

The rice flour used was leach rice flour (2016 “Asahi no Yume” from Tochigi prefecture; analyzed by the Japan Food Research Center: starch content, 77.8%; amylose content, 15.0%; protein content, 6.4%; starch damage, 2.3%; moisture content, 13.6%), milled by wet-airflow milling (Hinomoto Grain Flour Co., Ltd., Tochigi, Japan). The particle size distribution of the rice flour was determined. The median, average, minimum, and maximum diameters of the rice-flour particles were 34.3, 45.5, 0.7, and 251.7 μm, respectively. The particle size distribution showed two peaks at approximately 10 and 50 μm. Granulated sugar (Mitsui DM Sugar Co., Ltd., Tokyo, Japan), salt (The Salt Industry Center), olive oil (J-Oil Mills, Inc, Tokyo, Japan), and dry yeast (S.I. Lesaffre, Marcq-en-Barœul, France) were used as auxiliary ingredients. In addition, we used the thickening polysaccharides often used in processed foods: modified tamarind gum [32] (MTG (thermoresponsive xyloglucan); Sumitomo Pharma Food & Chemical Co., Ltd., Osaka, Japan), tamarind gum (TG; Sumitomo Pharma Food & Chemical Co., Ltd., Osaka, Japan), and xanthan gum (XT; Sumitomo Pharma Food & Chemical Co., Ltd., Osaka, Japan).

### 2.2. Sample Preparation

#### 2.2.1. Polysaccharide Thickener Solutions

The polysaccharide thickener solutions (0.25, 0.5, 0.75, and 1.0%) were prepared by adding a small quantity of each polysaccharide in the powder form to pure distilled water; TG and XT were dissolved at 25 °C, and MTG was dissolved in distilled water at 5 °C or lower.

#### 2.2.2. Bread

The basic recipe for rice-flour bread batter required 8 g of granulated sugar, 1.5 g of salt, 1.5 g of dry yeast, 3 g of olive oil, and 90 g of distilled water per 100 g of rice flour. The polysaccharide thickener solutions were mixed with the rice flour at concentrations ranging from 0.25% to 1.0%. The amount of distilled water added ranged from 92.5 g (0.25% polysaccharide) to 100 g (1.0% polysaccharide), considering the water absorption rate of the polysaccharide thickening agent. For the rice-flour bread batter, 100 g of rice flour and each polysaccharide thickener solution (or 90 g distilled water for the control) were mixed using a mixer (Aikoh Kenmix Aikoh Premier KMM770, Aikosha Mfg. Co., Ltd., Saitama, Japan) at 35 °C and 150 rpm for 1 min. Subsequently, the batter was allowed to absorb water at 25 °C for 29 min. Granulated sugar, salt, and dry yeast were added, and the batter was kneaded at 150 rpm for 1 min. Olive oil was then added, following which the batter was kneaded at 150 rpm for 1 min. Finally, the batter was prepared by kneading for another 10 min at 150 rpm.

The batter was first fermented for 30 min at 38 °C and 90% relative humidity (RH) in a whirlpool (QBX-132HRST1, Fukushima Galilei Co., Ltd., Osaka, Japan). It was then degassed and poured into a muffin mold (silicon, 5 cm in diameter at the bottom and 7 cm in diameter at the top) in 35 g portions using an electronic balance. This was followed by a second fermentation for 20 min under the same conditions mentioned. Subsequently, the rice-flour bread was baked in an oven (Combination Range DR504E, Harman Co., Ltd., Osaka, Japan) at 200 °C for 14 min. The bread was then cooled at 25 °C for 2 h, placed in a sealed polypropylene container, and stored at 4 °C and 65% RH for 0–3 days to be used for measurements.

### 2.3. Powder Characteristics of Rice Flour

#### 2.3.1. Water Absorption Characteristics

The water absorption and hydration rate of the rice flour were measured [33]. A glass filter paper (37 mm*ϕ*) and 7 mL of the sample were placed inside an aluminum cylinder (38 mm*ϕ*) with holes, immersed in water, and left for 1 h to obtain a water absorption–time curve. The moisture content (g-water/g-dry matter) at the point when no change in water absorption was observed was considered the saturated moisture content. The water absorption was obtained from the difference between the initial and measured moisture contents of the rice flour.

#### 2.3.2. Gelatinization Characteristics

The sample was prepared by combining the rice-flour sample with distilled water such that the sample concentration was 8% anhydrous equivalent and the total weight was 26 g. The stirring speed was 160 rpm. The pasting characteristics of rice flour were measured using the Rapid Visco-Analyzer (Model RVA-3D, Newport Scientific, Warriewood, Australia). The analyzer used a program wherein the temperature was held at 50 °C for 1 min, increased to 95 °C over 7 min and maintained for 5 min, and then decreased to 50 °C over 6 min. A curve was acquired for a total of 19 min to obtain the pasting start temperature, peak viscosity and temperature, breakdown, setback, and final viscosity [34].

#### 2.3.3. Scanning Electron Microscopy

The rice flour or polysaccharide thickener (powder) was fixed to a sample stand. Subsequently, Pt/Pd was deposited using an ion spatter (Model E-10130, Hitachi, Ltd., Tokyo, Japan) and observed using a tabletop scanning electron microscope (S-800, Hitachi, Ltd., Tokyo, Japan) at an accelerating voltage of 15 kV and a magnification of 150–4000 times.

### 2.4. Batter Characteristics

#### 2.4.1. Flow Characteristics (Apparent Viscosity)

The apparent viscosities of the polysaccharide thickener solution and prepared batter were measured using a cone-plate-type rotational viscometer (TV-20 type cone-plate viscometer TVE-20H, Toki Sangyo Co., Ltd., Tokyo, Japan). The value after 2 min was measured at 25 °C and rotational speeds of 3, 6, 12, 30, 60, and 100 rpm. The cone rotor measured 1°34′ × R24 and 3° × R12 for the polysaccharide thickener solution and bread batter, respectively.

#### 2.4.2. Dynamic Viscoelasticity

The dynamic viscoelasticities of the polysaccharide thickener solutions were measured using a Rheolograph Sol (SIC type, Toyo Seiki Co., Ltd., Tokyo, Japan) with an amplitude of ±100 μm and a frequency of 3 Hz. The temperature dependence was measured using a 45 min program wherein the temperature was held at 5 °C for 5 min, increased to 80 °C over 15 min and maintained for 5 min, and then decreased to 5 °C over 15 min [35].

The prepared rice-flour batter was measured using a dynamic viscoelasticity measuring device (ARES100FRT-N1: Rheometric Scientific Far East Ltd, Tokyo, Japan). A 25 mm diameter parallel plate fixture with a 1.0 mm gap was used. The strain dependence was measured at a frequency of 1.0 rad/s and 15 °C by varying the strain from 0.01% to 100% to identify the linear region. The temperature dependence was measured at a frequency of 1.0 rad/s and a strain rate of 0.1% between 15 and 50 °C. The dynamic viscoelasticities, storage modulus (G′), loss modulus (G″), and loss tangent (tanδ = G″/G′) were obtained [36].

### 2.5. Bread Characteristics

#### 2.5.1. Specific Volume

The bread was weighed after cooling at 25 °C. The apparent volume was measured by the rapeseed method [37], following which the specific volume was calculated.

#### 2.5.2. Moisture Content of Bread

A sample (weighing approximately 3 g) from the crumb of the rice-flour bread was used to determine the moisture content using a halogen moisture meter (HB43-S: METTLER TOLEDO Co., Ltd., Tokyo, Japan). The measurement was based on the weight change over 120 s and was completed when the weight change was within 1 mg/s [38].

The ratio of the moisture content on the day of measurement to that on the day of baking was determined to evaluate the moisture retention of bread.

#### 2.5.3. Mechanical Properties of Bread

Cubes (20 mm × 20 mm × 20 mm) were cut from the crumb of the bread, and their rupture properties were measured using a creep meter (Rheoner RE-3305S, Yamaden Co., Ltd., Tokyo, Japan). An acrylic resin disk plunger with a diameter of 40 mm was used to compress the sample to 99% of its height at a compression speed of 6 cm/min. Stresses at 25%, 40%, and 70% compression [38] were obtained from the stress–strain curve, and the hardness of the bread was evaluated.

#### 2.5.4. Bread Color

Cubes (20 mm × 20 mm × 20 mm) were cut from the crumb of the bread. The color was measured using the CIE L*a*b* color space [26,39] and a color spectrophotometer (ND-1001DP, Nippon Denshoku Kogyo Co., Ltd., Tokyo, Japan). The L* (lightness), a* (redness), and b* (yellowness) values were determined to evaluate the color of the crumb of the bread, by measuring three points in the center of the sample (*n* = 3–7). The gluten-free rice-flour bread is semi-infinite solid, so the measurement conditions were as follows: light source, halogen lamp (D65); detector, photodiode; standard observer, 2°; measurement diameter, 10 mm*ϕ*(Compliant with JIS Z8722).

#### 2.5.5. Sensory Evaluation

Sensory evaluation was conducted to assess the palatability of gluten-free rice-flour bread with a polysaccharide thickener. The samples were obtained after the bread was baked, cooled at 25 °C, placed in an airtight polypropylene container, and stored at 4 °C for 1 h.

The sample was a muffin-shaped bread (7 cm in diameter at the top and 5 cm in diameter at the bottom) radially divided into 6 equal pieces, with each piece consisting of 3 slices. The 3 types of rice-flour bread with each polysaccharide thickening agent (1.0% additive of MTG, TG, or XT) were evaluated on a 7-point scale by a panel of 30 untrained female university students, with a control (no additive) as the standard. They were analytically evaluated based on the surface browning and gloss, texture, aroma when placed in the mouth, softness, stickiness, moistness, and residual feeling. Additionally, they were tested for preference based on the aroma when placed in the mouth, sweetness, aftertaste, and overall experience.

#### 2.5.6. Retrogradation Characteristics of Bread

The control and polysaccharide-thickened rice-flour breads were stored at 25 °C and 65% RH for 0–3 days. Subsequently, approximately 20 g of the crumb was ground with a mortar while dehydrating with approximately 60 mL ethanol and then filtered. This process was repeated twice. The sample was then defatted with acetone and air dried to obtain a defatted powder sample [40]. The defatted powder sample (150 mg) was placed in a 1 cm diameter mold and compressed at 60 N for 3 min using a hydraulic pump (RIKEN POWER, RIKEN KIKI Co., Ltd., Tokyo, Japan) to form a tablet-shaped sample. The X-ray diffraction intensities were measured with the tablet samples using an X-ray diffractometer (RINT-1500, Rigaku Electric Co., Ltd., Tokyo, Japan) with a Cu tube sphere and the following parameters: diffraction angle 2θ = 4°–40°, voltage = 36 kV, current = 150 mA, and scanning speed = 2°/min. The samples were irradiated, and the state of crystallinity recovery was observed from the diffraction intensity [39]. As retrogradation starch is known to exhibit a sharp peak at a diffraction angle of 17°, the intensity at 2θ = 17° was considered an indicator of retrogradation [41].

### 2.6. Statistical Analysis

Statistical processing of the measured objective characteristics of the samples and sensory evaluation results was conducted using the statistical analysis software Excel Statistics 2012. Further, one-way analysis of variance was performed, and the samples that exhibited significant differences were further tested by multiple comparisons using Scheffe’s method. The significance level for all tests was set to 5%.

### 2.7. Ethical Considerations

The purpose of this study was explained to the panel, and their understanding and consent were obtained before conducting the sensory evaluation. This study was carried out with the approval of the “Japan Women’s University Ethical Review Committee for Experimental Research on Human Subjects” (Ethics Committee Approval No. 279).

## 3. Results and Discussion

### 3.1. Water Absorption Characteristics

The water absorption characteristic curve of rice flour is shown in Figure 1. The saturated water absorption of rice flour reached 0.76 g in 8 min, and half the saturated water amount was absorbed in 77 s (the water absorption rate). Matsuki et al. compared the water absorption of rice flour milled by four different milling methods and reported that wet-airflow-milled rice flour had the highest water absorption rate and lowest water absorption [33]. The water absorption and water absorption rate of the rice flour used in this study were almost equal to those of the rice flour milled by wet-airflow milling in the study by Matsuki et al. Shoji et al. also reported the possibility that the water absorption of rice flour is a slow process, in which water absorption by capillary attraction occurs first, followed by penetration and diffusion. During penetration into the rice flour, the water on the surface of the rice flour that penetrates the voids causes the starch to swell. In rice flour with a high degree of starch damage, the swollen starch area resists water movement, consequently reducing the water absorption rate. The starch damage level of the rice flour used in this study was lower than that of wet-airflow-milled rice flour, suggesting that the water absorption rate was higher [33].

### 3.2. Gelatinization Characteristics

The RVA curve of rice flour is shown in Figure 2, and the gelatinization characteristics are shown in Table 1. The rice flour used in this study had a gelatinization temperature of 71 °C, which is higher than the upper limit of the reported gelatinization temperature range of 60.6–64.3 °C for low-amylose rice [42]. However, the gelatinization temperatures of high- and medium-amylose rice have been reported to be 64.8–65.2 °C and 66.0–80.1 °C, respectively [43], indicating that the prepared rice flour is similar to that from medium-amylose rice.

### 3.3. Scanning Electron Micrographs

The scanning electron micrographs of rice flour are shown in Figure 3. The particle size distribution of the rice flour used in this study revealed a mixture of small and large grain sizes; therefore, observations were made separately for grain sizes of <25 µm and >25 µm.

Particle sizes below 25 µm had a compound grain structure with small aggregates on the surface, whereas those larger than 25 µm had fewer small aggregates and a smooth surface.

The flour particles exhibited a compound grain structure. It has been reported that rice flour milled by wet-airflow milling from the amyloplasts in cellular tissues has a starch grain structure [44]. This study revealed an uneven surface at small grain sizes for the aforementioned reasons.

Furthermore, the scanning electron micrographs of the polysaccharide thickeners used in this study are shown in Figure 4. MTG had larger particles with smoother surfaces compared with those of TG. The particles of XT were smaller and more angular in shape than those of TG, indicating that, although both MTG and TG originated from tamarind seed gum, they possessed different microstructures.

### 3.4. Properties of Polysaccharide Thickener Solutions

#### 3.4.1. Flow Characteristics

Figure 5 shows the shear rates of the polysaccharide thickener solutions, which exhibited nearly Newtonian fluid behavior in MTG and TG. The apparent viscosity increased with increasing concentration. At 25 °C, the apparent viscosity of XT decreased with increasing shear rate and indicated shear-thinning flow, which was not significantly affected by concentrations of >0.5%. These results are consistent with previously reported results [45,46].

#### 3.4.2. Dynamic Viscoelasticity

The storage modulus (G′) and loss tangent (tanδ) in the dynamic viscoelasticity of the polysaccharide thickener solutions are shown in Figure 6.

The storage modulus of MTG increased when the temperature decreased from 80 to 40 °C but decreased upon cooling from approximately 40 to 5 °C. All concentrations of TG showed a constant storage modulus with increasing temperature and little change with decreasing temperature, while the storage modulus of XT increased with increasing concentration and decreasing temperature from 80 to 5 °C.

The loss tangent, which represents the ratio of the loss modulus to the storage modulus, increased for MTG as the temperature decreased from 40 to 5 °C, with the viscous component particularly increasing at temperatures lower than 35 °C. The storage modulus increased as the temperature decreased from 75 to 40 °C, while the loss modulus increased from approximately 40 °C. The gelatinization of MTG at ~40 °C and its tendency to become a sol at temperatures below 40 °C were confirmed.

It was also found that a concentration of more than 0.75% was required for the development of viscoelasticity in MTG. TG showed little change in both storage and loss modulus; hence, the change in behavior was also small for the loss tangent, but the viscous factor increased below 20 °C. XT showed that the loss tangent was almost constant irrespective of concentration and was shown to be less affected by temperature.

### 3.5. Batter Characteristics

#### 3.5.1. Flow Characteristics

Figure 7 shows the apparent viscosity of the batter with each polysaccharide solution. The apparent viscosity of the MTG-added batter increased with increasing concentration but was less sensitive to concentration at higher shear rates. By contrast, the apparent viscosity of the TG-added batter was lower than that of the control at 0.25% addition. Similar to that of the MTG-added batter, the apparent viscosity of the XT-added batter increased with increasing concentration, but the trend reversed at a shear rate of 60 s^−1^.

#### 3.5.2. Dynamic Viscoelasticity

The temperature dependence of the dynamic viscoelasticity of the batter with each polysaccharide thickener is shown in Figure 8. The storage modulus was lower for all the polysaccharide thickeners compared to those of the control. The storage modulus of the MTG-added batter decreased with increasing concentration, but the opposite trend was observed for the TG-added batter. The storage modulus of the XT-added batter was almost unaffected by changes in its concentration. The tanδ of the MTG-added batter increased above 35 °C but was lower than that of the other polysaccharide thickeners, indicating a larger storage modulus. The TG- and XT-added batters showed a similar trend. Maninder reported that tamarind seed gum underwent a large viscosity breakdown, with a minor drawback: it was sticky but not retrogradation-resistant [47]. The viscosity of the solution may have influenced these properties. However, the MTG-added batter exhibited a lower tanδ and larger elastic modulus than did the TG- and XT-added batters, regardless of the additive concentration. This suggests that the MTG solution tends to gel at approximately 40 °C and that this property is also reflected in the dynamic viscoelasticity of the batter.

These results indicate that the basic properties of the polysaccharide thickeners affect the characteristics of the batter: the TG system had a low viscosity close to that of the control when added to the batter, while the XT system had a high viscosity even at low concentrations. MTG was gelatinized at high temperatures and solubilized at low temperatures, suggesting that its sol–gel transition point is 35–40 °C. Reflecting this property, when MTG was added to the batter, the tanδ decreased at temperatures above 30 °C and the elastic factor increased, suggesting that the properties of the solution affected the batter.

### 3.6. Bread-Making Properties

#### 3.6.1. Specific Volume and Moisture Content of Bread

The specific volume and moisture content of gluten-free rice-flour bread baked with the polysaccharide thickeners are shown in Figure 9. The specific volume was lower for bread with 0.25% polysaccharide thickeners added than for bread with no additives (hereafter referred to as control) (*p* < 0.05); the specific volume of bread with MTG and TG tended to be smaller than tha of bread with XT added. This is likely because the addition of polysaccharides increased the viscosity of the batter and prevented it from swelling.

The moisture content was approximately 40% in the control. In contrast, the amount of water added increased as the quantity of polysaccharides added increased for the MTG-, TG-, and XT-added breads (*p* < 0.05), indicating that the water retention improved. This result suggests that the ability of the polysaccharide thickeners to prevent water release also had an effect on the bread by increasing the moisture content. The water-soluble polysaccharides used contain several hydroxyl groups, which improved water retention. TG has been reported to possess strong hydration and excellent water-retention properties [48]. Compared to the XT-added bread, the TG-added bread showed superior water retention, which was influenced by the molecular structure of the polysaccharide thickeners.

#### 3.6.2. Mechanical Properties of Bread

The stresses at 25%, 40%, and 70% compression for the gluten-free rice-flour bread baked with polysaccharide thickeners are shown in Figure 10. The 0.5% MTG- and 0.25% TG-added breads showed the lowest stress at 25% compression (*p* < 0.05). However, at 70% and 25% compression, the stress of the MTG-added bread decreased with increasing MTG concentration (*p* < 0.05), while the TG-added bread showed the lowest stress at 0.25% (*p* < 0.05).

Changing the concentration was effective in improving the overall bread quality. However, the surface hardness of the bread was not affected by the addition of XT at any concentration, while the internal phase tended to be the hardest at 0.5% addition.

#### 3.6.3. Bread Color

The chromaticity on the surface of the gluten-free rice-flour bread baked with the polysaccharide thickeners is shown in Figure 11. The L* (lightness) value was slightly higher than that of the control for all polysaccharide thickener additions (*p* < 0.05), but there was no significant difference between the concentrations. The a* (redness) and b* (yellowness) values decreased with increasing polysaccharide concentration in the MTG-added bread (*p* < 0.05), but increased for the 0.25% TG-added bread and decreased at higher concentrations of TG (*p* < 0.05). The b* values of the XT-added bread decreased at 0.25% and 0.5% additions (*p* < 0.05), showing a different trend from that of the TG-added bread.

The color of the bread showed a tendency for redness and yellowness to decrease with increasing concentrations of TG and MTG additions, which differed for XT. Regarding the color of gluten-free rice-flour bread, it has been reported that the L and a values are lower when the specific volume of the bread is lower [23]. This factor is thought to be related to the retention of air bubbles during baking and the texture of the bread. In this study, as shown in Figure 9, the specific volume of the bread decreased with the addition of thickened polysaccharides, suggesting that the lower quantity of air bubbles in the bread affected the lower a value. Additionally, the amino-carbonyl reaction between rice-flour protein and added sugar during baking was suppressed at higher concentrations of thickened polysaccharides, which may be due to the increased dough viscosity and different properties of the thickened polysaccharides.

#### 3.6.4. Sensory Evaluation

Sensory evaluation was conducted using the grading method to evaluate the palatability of rice-flour bread with added polysaccharides. Samples with 0.5% addition of either the MTG, TG, or XT polysaccharide thickener and bread without additives were evaluated. Higher scores of the samples were considered to render them superior to the control for all parameters except the baking color and flavor (*p* < 0.05). The results are shown in Figure 12. The MTG-added bread was evaluated to be lighter in baking color than the control (*p* < 0.05), and the XT-added bread was evaluated to be finer in grain than the control (*p* < 0.05). Furthermore, the additives were expected to mask the flavor of rice flour. The flavor of the TG-added bread was significantly weaker than that of the control (*p* < 0.05), but the flavor of the MTG- and XT-added bread was similar to that of the control. The XT-added bread was the softest, followed by the MTG-added bread (*p* < 0.05). No significant difference in softness were found between the TG-added bread and the control. The moistness of the bread was evaluated to be greater than that of the control for all polysaccharide thickeners (*p* < 0.05). Additionally, the MTG-added bread was preferred over the control in terms of sweetness (*p* < 0.05), aftertaste (*p* < 0.05), and overall preference evaluation (*p* < 0.05).

### 3.7. Preservation of Gluten-Free Rice-Flour Bread

#### 3.7.1. Moisture Content

The effect of storage on the moisture content of gluten-free rice-flour bread with added polysaccharides is shown in Figure 13. The moisture content tended to increase with the addition of polysaccharide thickeners on the day of baking compared to that of the control. The addition of thickener tended to result in a higher value than that of the control up to the first day, but on the third day, only the XT-added bread showed a moisture-retention effect. However, at polysaccharide thickener concentrations of 0.5% or higher, all polysaccharide thickeners showed improved moisture retention compared with that of the control. MTG and TG have many hydroxyl groups and hydrophilic side chains that bind strongly to water [48], making them effective in increasing the moisture content. Furthermore, TG and XT have small particles and large surface areas, while MTG has large particles and a small surface area, suggesting that this may influence water retention.

#### 3.7.2. Mechanical Properties

Figure 14 shows the effect of storage on the stress at 25% and 70% compression of the gluten-free rice-flour bread with the polysaccharide thickener. The 25% compression stress of the bread with MTG was lower than that of the control until the first day at 0.25% and 0.5% MTG concentrations. However, the value increased rapidly on the second day (*p* < 0.05). The MTG-added bread was softer than the control until day 2 at 0.5% MTG concentration and until day 1 at 0.75% MTG concentration, indicating that the addition of MTG inhibited the hardening of the inner phase of the bread. The TG-added bread showed the lowest value at 0.5% TG concentration, which suppressed the hardening of the bread. Furthermore, the XT-added bread showed a smaller value at 1.0% XT concentration than did the control and TG-added bread, which suppressed the hardening of the bread. The values of the XT-added bread indicated that higher concentrations have a greater effect on hardening inhibition throughout the 3-day storage period.

#### 3.7.3. Retrogradation Characteristics of Bread

To investigate whether retrogradation caused the hardening of the bread, the relative intensities of gluten-free rice-flour bread with 0.5% polysaccharide thickener were calculated from the X-ray diffraction pattern (at 2θ = 17°; Figure 15) and are shown in Figure 16. Compared with the control, the bread with MTG showed lower relative intensities after 2 days. Similar to those of the MTG-added bread, the relative intensities of the TG-added bread were low from the second day onward. The relative intensity of the XT-added bread was higher than those of the MTG- and TG-added breads and near that of the control. This indicates that at 0.5% concentration, the addition of XT to bread has no inhibitory effect on starch retrogradation. It has been reported that TG inhibits the starch retrogradation caused by the rearrangement of amylose and amylopectin by entanglement with starch molecules [49]. The same rationale for the inhibitory effect on starch retrogradation was considered to be true in this study.

### 3.8. Relationship between the Objective Characteristic Values and Sensory Evaluation Values of Gluten-Free Rice-Flour Bread

A positive correlation was observed between the apparent viscosity of the batter and the stress on the bread at 70% compression at TG and XT (Figure 17). Specifically, unlike that for the other polysaccharide-added breads, the apparent viscosity of the batter differed significantly for the TG-added bread, depending on the concentration of TG.

A positive correlation was obtained between the batter viscosity and the softness of the bread in the sensory evaluation, indicating that the higher the batter viscosity, the softer the bread (*p* < 0.05) and the finer the grain texture (*p* < 0.1) (Figure 18).

Furthermore, the moisture content affected the moistness and softness of the bread in the sensory evaluation (*p* < 0.01). A higher moisture content indicated that the bread was softer and moister (*p* < 0.1) (Figure 19).

These results indicate that the type and concentration of polysaccharide thickeners added to gluten-free rice-flour bread have different effects on their bread-making and preservation properties. In terms of shelf life, the addition of 0.5–0.75% of polysaccharides inhibited the hardening and retrogradation of MTG-added bread. In comparison, the addition of TG to bread at a lower concentration of 0.5% had an inhibitory effect on hardening at 25% compressive strength, while XT-added breads required a 1.0% concentration.

The addition of MTG improved the bread palatability and the bread-making and shelf-life properties of gluten-free rice-flour bread. In addition, a positive correlation was observed between the viscosity coefficient of the batter and the texture and softness of the bread, suggesting that the viscosity coefficient of the batter is important for controlling the texture and softness of gluten-free rice-flour bread. Sensory evaluations revealed that adding polysaccharide thickeners improved the overall consumption experience owing to their various effects, such as increasing the moisture content of the breads. It was suggested that the viscosity coefficient of the batter can be used to control the grain texture and softness of gluten-free rice-flour bread, and the quality of the final product could be regulated by controlling the composition of the batter.

## 4. Conclusions

Gluten-free rice-flour bread hardens over time and possesses poor storage qualities, which are attributed to its short shelf life of only 3 days. In this study, the type and concentration of the added polysaccharide thickener were found to exert different effects on the bread-making properties and shelf life. Results indicated that the addition of 0.5% MTG/TG or 1% XT effectively prevented hardening. Further, the addition of TG or MTG to gluten-free rice-flour bread was more effective in bread making and preservation at lower concentrations than the addition of XT. MTG addition reduced the specific volume of the bread but increased the moisture content and retention. It also reduced the stress at 70% compression and was found to be soft, moist, and preferred over the other additives and control, demonstrating the positive impact of the thermoresponsive xyloglucan additive. Further investigations into the bread-making properties of gluten-free rice-flour bread are essential for improving preferable qualities and preservation characteristics, as they could greatly help people suffering from wheat-flour and other allergies, as well as celiac disease, by providing good alternative breads and baked goods based on rice flour.

Further, we believe that this paper will be of interest to the readership of this journal because people suffer from wheat-flour and other allergies.

## Figures and Tables

**Figure 1 foods-12-02761-f001:**
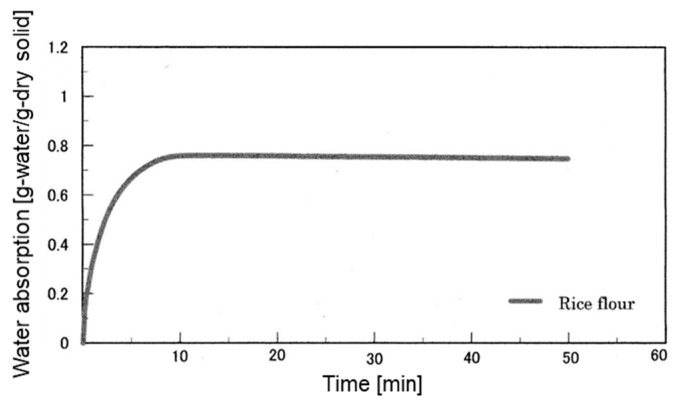
Water absorption curve of rice flour.

**Figure 2 foods-12-02761-f002:**
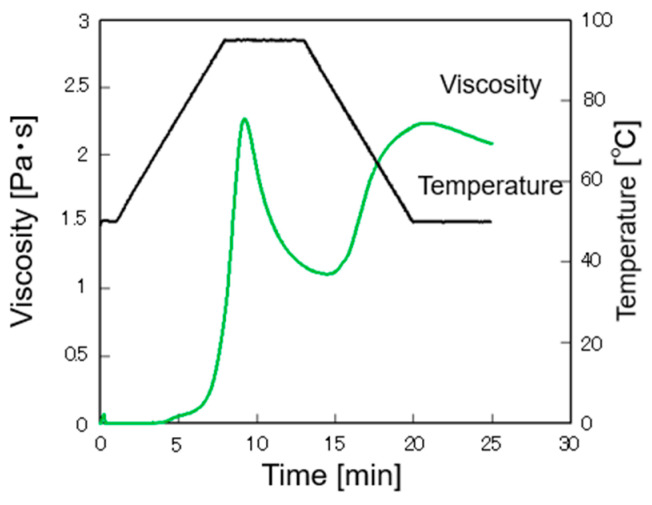
RVA viscosity curve of rice flour.

**Figure 3 foods-12-02761-f003:**
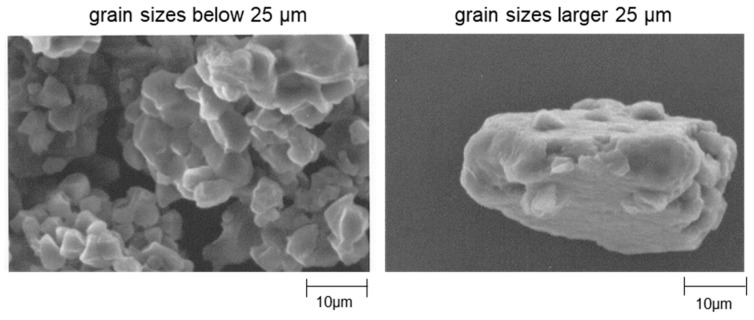
SEM micrographs of rice flour.

**Figure 4 foods-12-02761-f004:**
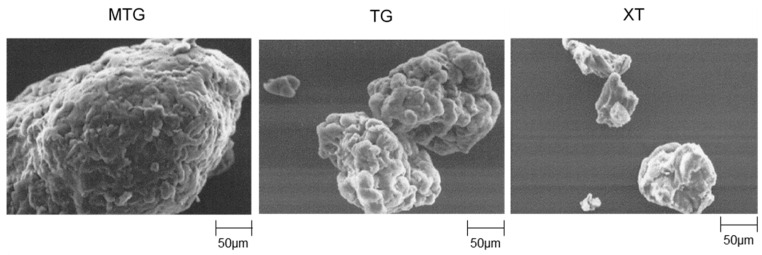
SEM micrographs of polysaccharide thickeners.

**Figure 5 foods-12-02761-f005:**
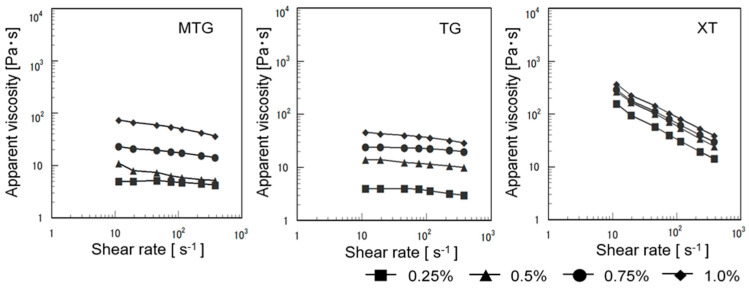
Apparent viscosity of polysaccharide thickener solutions.

**Figure 6 foods-12-02761-f006:**
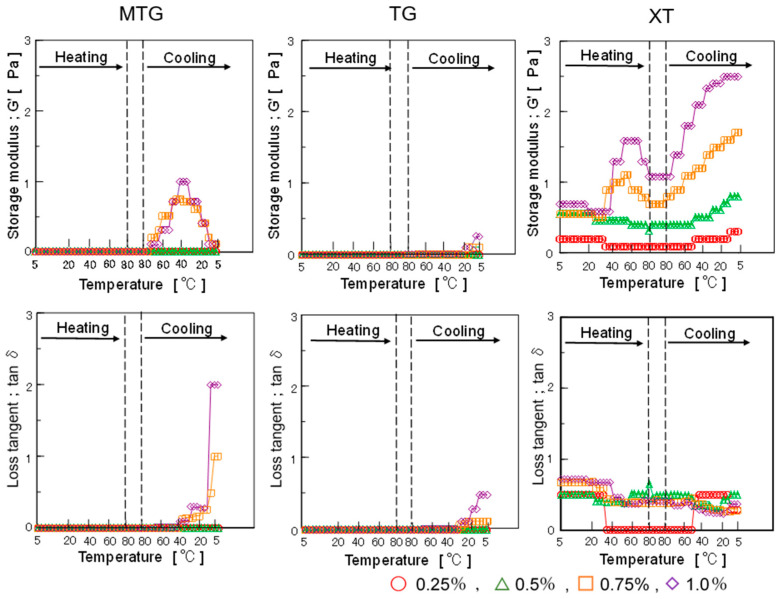
Effect of temperature on storage modulus (G′) and loss tangent (tanδ) of polysaccharide thickener solutions.

**Figure 7 foods-12-02761-f007:**
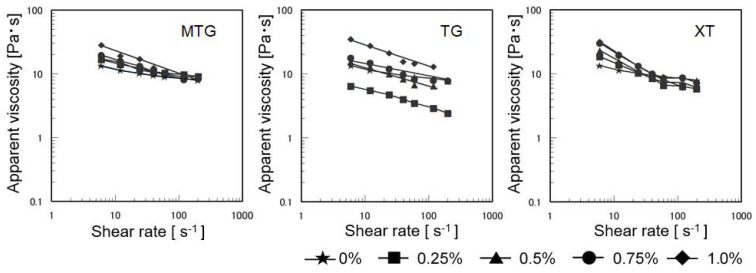
Apparent viscosity of batter with each polysaccharide solution at 25 °C.

**Figure 8 foods-12-02761-f008:**
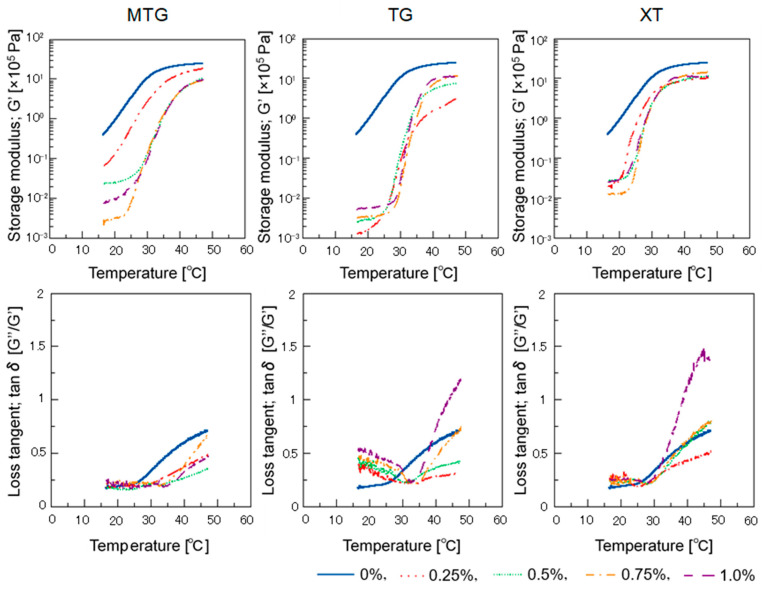
Effect of temperature on storage modulus (G′) and loss tangent (tanδ) of batter with polysaccharide thickeners.

**Figure 9 foods-12-02761-f009:**
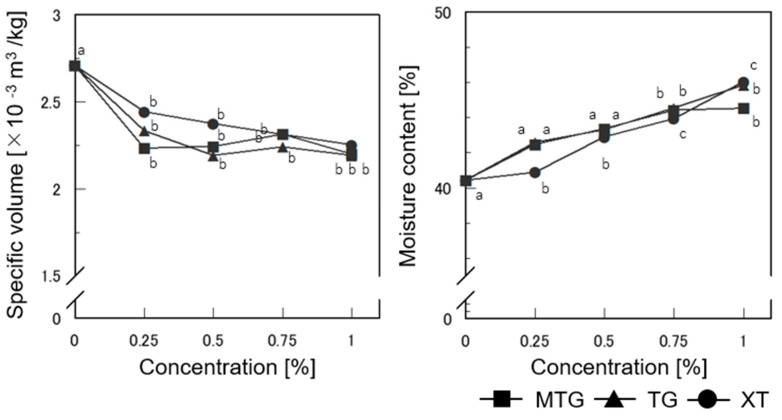
Effect of polysaccharide thickeners on specific volume and moisture content of gluten-free rice-flour breads. a–c Different letters indicate significant difference between concentrations (*n* = 3; *p* < 0.05). Means indicated by different letters are significantly different at *p* < 0.05 according to Scheffe’s multiple range test.

**Figure 10 foods-12-02761-f010:**
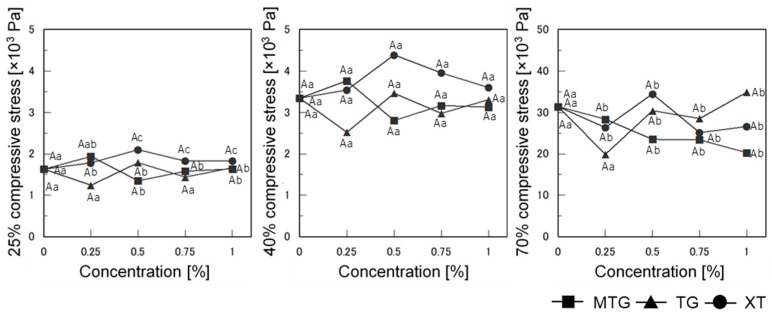
Effect of polysaccharide thickeners on 25%, 40%, and 70% compressive stress of gluten-free rice-flour breads. A letter indicates that there is no significant difference between the samples (*n* = 3; *p* < 0.05). a–c Different letters indicate significant difference between concentrations (*n* = 3; *p* < 0.05). Means indicated by different letters are significantly different at *p* < 0.05 according to Scheffe’s multiple range test.

**Figure 11 foods-12-02761-f011:**
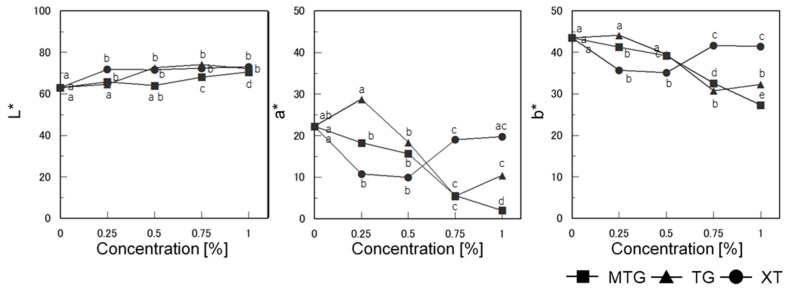
The color on the surface of the gluten-free rice-flour breads. a–e Different letters indicate significant difference between concentrations (*n* = 3; *p* < 0.05). Means indicated by different letters are significantly different at *p* < 0.05 according to Scheffe’s multiple range test.

**Figure 12 foods-12-02761-f012:**
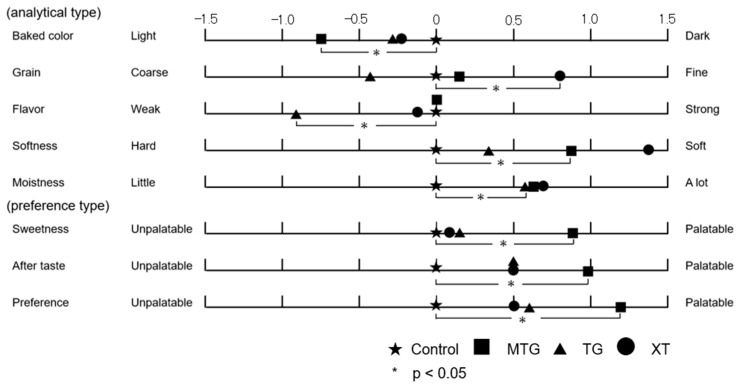
Sensory evaluation results of gluten-free rice-flour breads with polysaccharide thickeners at 0.5%. * indicates closest significant difference from the control at *p* < 0.05; higher values are also significantly different to the control.

**Figure 13 foods-12-02761-f013:**
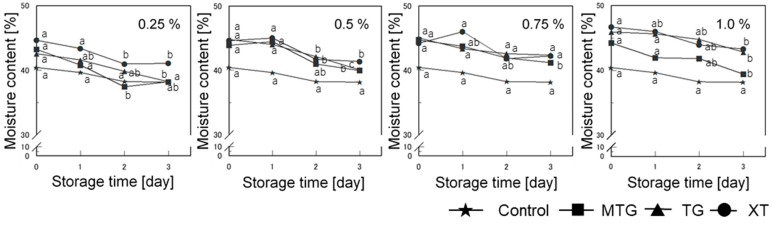
Effect of storage on moisture content of gluten-free rice-flour breads with polysaccharide thickeners. a–c Different letters indicate significant differences between storage times (*n* = 3; *p* < 0.05). Means indicated by different letters are significantly different at *p* < 0.05 according to Scheffe’s multiple range test.

**Figure 14 foods-12-02761-f014:**
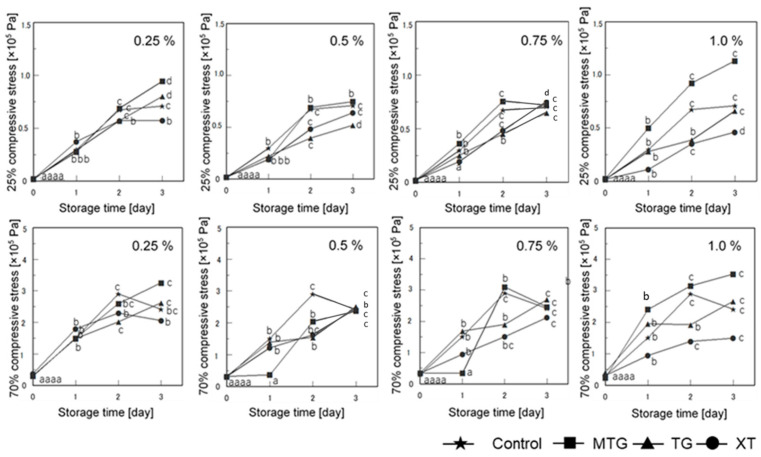
Effect of storage on 25% and 70% compressive stress of gluten-free rice-flour bread. (Figure 3. *p* < 0.05). Means indicated by different letters are significantly different at *p* < 0.05 according to Scheffe’s multiple range test.

**Figure 15 foods-12-02761-f015:**
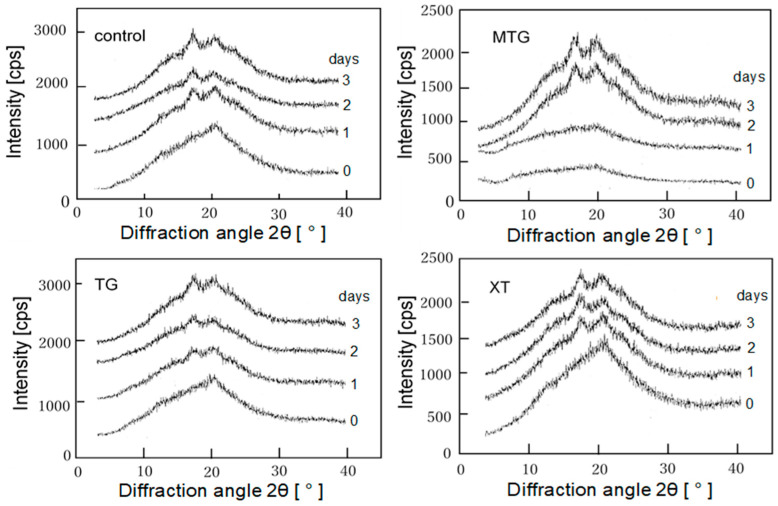
X-ray diffraction diagram of gluten-free rice-flour breads with 0.5% polysaccharide thickeners.

**Figure 16 foods-12-02761-f016:**
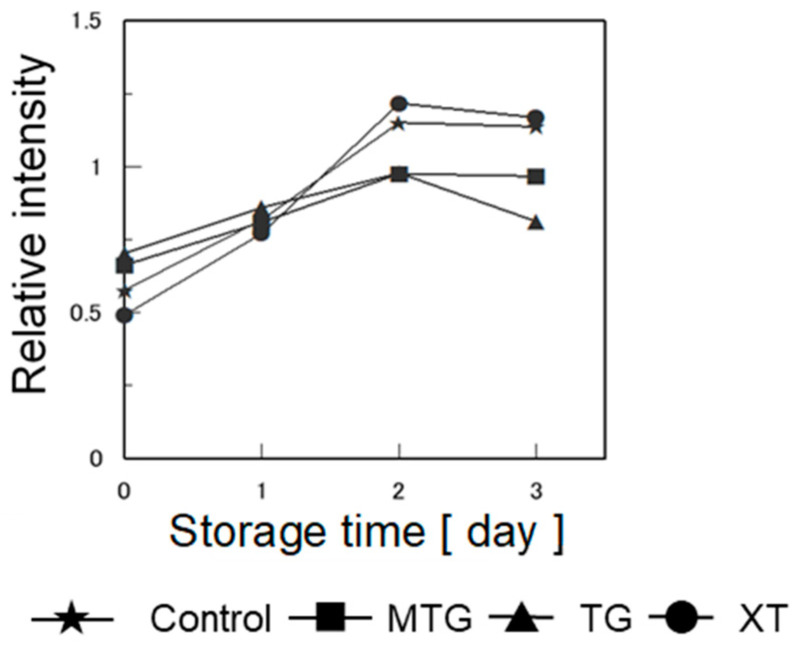
Effect of storage on relative intensity of X-ray diffraction at 2θ = 17° in gluten-free rice-flour bread with 0.5% polysaccharide thickeners. Relative intensity: 17° intensity/20° intensity.

**Figure 17 foods-12-02761-f017:**
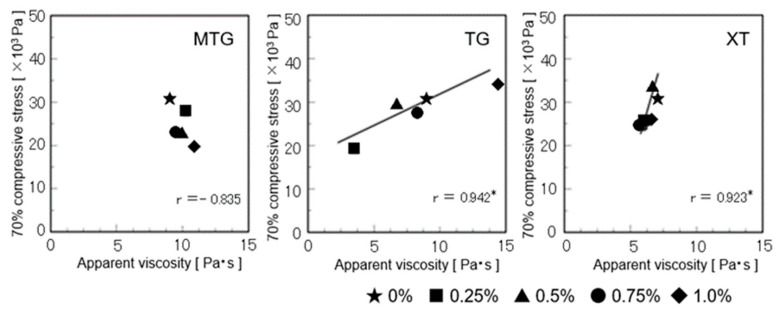
Relationship between the batter’s apparent viscosity and the 70% compressive stress of gluten-free rice-flour breads (baking date). * Correlation is significant at *p* < 0.05.

**Figure 18 foods-12-02761-f018:**
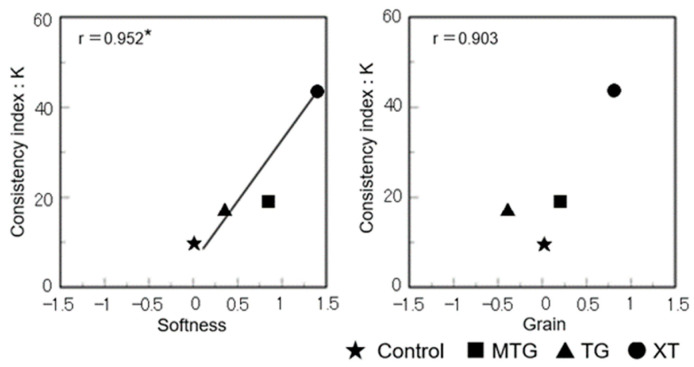
Relationship between the batter’s flow characteristics and sensory evaluation results of gluten-free rice-flour breads. * Correlation is significant at *p* < 0.05.

**Figure 19 foods-12-02761-f019:**
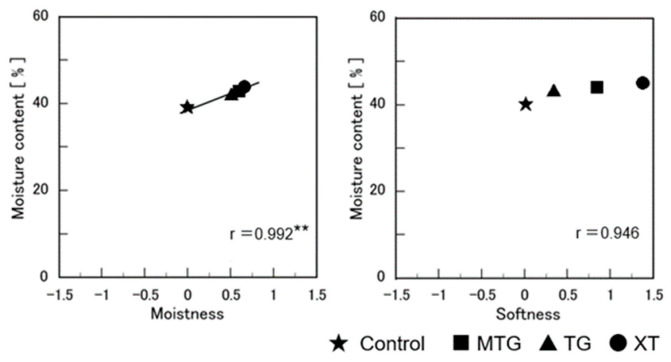
Relationship between water content and sensory evaluation results of gluten-free rice-flour breads. ** Correlation is significant at *p* < 0.01.

**Table 1 foods-12-02761-t001:** Gelatinization characteristics of rice flour.

		Rice Flour
Gelatinization temperature	[°C]	69.8
Peak viscosity	[Pa·s]	2.30
Minimum viscosity	[Pa·s]	1.28
Final viscosity	[Pa·s]	2.28
Break down	[Pa·s]	1.03
Set back	[Pa·s]	1.01

## Data Availability

Data are contained within the article.

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
