# Peer review of "Effect of Thermoresponsive Xyloglucan on the Bread-Making Properties and Preservation of Gluten-Free Rice-Flour Bread"

_foods, 2023, doi:10.3390/foods12142761_

Round 1
Reviewer 1 Report
Comments and Suggestions for Authors
The main objective of the present study was to investigate the effect of adding thermoresponsive xyloglucan on the bread-making properties and preservation of gluten-free rice-flour bread.
The manuscript in its present form contains many shortcomings. There are some examples:
Abstract: Please explain this statement. Which characteristics of bread determine MTG as the most preferred additive since the presence of MTG reduces its specific volume?
The last sentence is confusing.
INTRODUCTION: The introduction should be extensively rewritten to provide a clear overview of the topic and a clear rationale for the study. Since the employment of thermoresponsive xyloglucan is mostly widespread in other branches of the scientific community, such as tissue engineering and biomedicine, the question arises as to the justification of its addition to bakery products. A more complete and detailed literature review related to thermoresponsive xyloglucan properties and its application in food/bakery products should be included in the introduction. Moreover, the introduction needs to explain what the current gap in gluten-free bread research is.
"Unfortunately, these and cooking techniques that……." - this sentence is unclear.
"Additionally, gluten-free rice-flour bread could be prepared using ten types of rice flour by optimizing the amount of water added depending on the protein content of the rice flour [21]".- Ten types of rice flour? In the cited paper, the emphasis is not on the fact that bread can be made from the flour of 10 varieties of rice, but on the fact that the protein content must be taken into account as an indicator to adjust the quantity of water that will be suitable for breadmaking.
The aim of the study should be expressed in a more complex manner and more concisely, incorporating all aspects of the study.
Materials and methods/Results and Discussion:
The methodological approach is not clear enough to fully understand the experimental design and results.
Regarding the characteristics of the rice flour—composition, degree of starch damage, and particle size—there is no need to discuss these results since the aim of the study was to investigate the effect of polysaccharide thickeners on dough and bread properties. This result, together with the used methodology, should be moved to the Materials section.
Why were the water absorption and gelation characteristics determined only on the rice flour and not on the rice flour with added polysaccharide thickeners? What would be the relevance of these results to the aim of the study?
2.2.2. Bread: From the present method description, the procedure of dough preparation with added polysaccharide thickener solutions is not visible.
2.5.5. Sensory evaluation: The description should include all aspects of sensory analysis (trained or semi-trained panel, sample handling, conditions, etc.).
There is a lack of statistical information in the figures/tables to indicate whether the variations are significant or not. There is also no information on the number of analyses performed to obtain an analytical result.
The discussion is poorly written; it must be improved and particularly placed in the context of previously published papers.
All in all, the quality of the present manuscript is not satisfactory.
Author Response
Dear Editors and Reviewers
Thank you very much for reviewing our manuscript and offering your valuable advice.
We have addressed your comments with point-by-point responses, and revised the manuscript accordingly.
Responses to the Comments by Reviewer 1:
1.Abstract: Please explain this statement. Which characteristics of bread determine MTG as the most preferred additive since the presence of MTG reduces its specific volume?
The last sentence is confusing.
Reply:
Thank you for your suggestion. Although the specific volume of bread with MTG decreased, it was considered soft, moist, and tasty in sensory evaluations, which may be because of its high moisture content and low stress at 70% compression (indicating softness). This suggests that " The addition of MTG to gluten-free rice-flour bread reduced the specific volume, increased the moisture content, and reduced the stress at 70% compression. Therefore, the bread with MTG added was soft, moist, and preferred over other those with other additives."(Abstract L8-11). The last sentence was deleted and replaced with "We found that the thickening polysaccharides had to be added in appropriate concentrations to improve the bread-making properties and achieve the preferred effect.” (Abstract L.13-15).
2.INTRODUCTION: The introduction should be extensively rewritten to provide a clear overview of the topic and a clear rationale for the study. Since the employment of thermoresponsive xyloglucan is mostly widespread in other branches of the scientific community, such as tissue engineering and biomedicine, the question arises as to the justification of its addition to bakery products. A more complete and detailed literature review related to thermoresponsive xyloglucan properties and its application in food/bakery products should be included in the introduction. Moreover, the introduction needs to explain what the current gap in gluten-free bread research is.
Reply:
Thank you for the suggestion. Since there have been many reports on thermoresponsive xyloglucan in the fields of tissue engineering and biomedicine, but few reports in the food field, " A modified tamarind gum (thermoresponsive xyloglucan) has been widely reported in the fields of tissue engineering and biomedicine, including novel alternatives for tissue regeneration[25, 27] and temperature-controlled desorption of cells[28]. Until now, there have been several reports on quality improvement using polysaccharide thickeners in gluten-free rice-flour bread. However, there are only a few specific reports on a modified tamarind gum (thermoresponsive xyloglucan)."(Body P.2 L.1-6) and added related papers [25],[27],[28].
3."Unfortunately, these and cooking techniques that……." - this sentence is unclear.
Reply:
Thank you for your suggestion. I have corrected “Unfortunately, these breads are unsuitable for wheat allergy sufferers [18,19]. Hence, alternative ingredients and cooking techniques that can impart the effects of gluten are needed for further improving gluten-free breads.” to “Measures for controlling aging are similar, as they involve adding gluten; however, the resulting foods are inappropriate for consumption by people with wheat allergies [18,19].” in the text(Body P.1 L.26–27).
4. "Additionally, gluten-free rice-flour bread could be prepared using ten types of rice flour by optimizing the amount of water added depending on the protein content of the rice flour [21]".- Ten types of rice flour? In the cited paper, the emphasis is not on the fact that bread can be made from the flour of 10 varieties of rice, but on the fact that the protein content must be taken into account as an indicator to adjust the quantity of water that will be suitable for breadmaking.
Reply:
You have raised an important point and the sentence has been adapted: “Additionally, gluten-free rice-flour bread could be prepared using ten types of rice flour by optimizing the amount of water added based on the protein content of the rice flour [21].” To show the emphasis of the paper itself. However, optimizing the water added to protein content is outside the scope of our paper; and previous studies have often added wheat-derived additives in the preparation of rice-flour bread, but the authors have reported on the bread-making properties of 100% rice-flour gluten-free sponge cakes and breads that do not contain wheat derivatives. In the present study, we consider it worthwhile to insert this literature in order to report on the bread-making properties of 100% rice flour gluten-free breads to which non-allergenic thickened polysaccharides were added.
5.The aim of the study should be expressed in a more complex manner and more concisely, incorporating all aspects of the study.
Reply:
Thank you for your clarifying response; we have adapted the aims statement to: “Therefore, this study aimed to examine the effects of a modified tamarind gum (thermoresponsive xyloglucan) on the bread-making properties and preservation of gluten-free rice-flour bread, as well as the impact on the sensory qualities of the alternative bread.” (Body P.2 L.6–8)
6.Materials and methods/Results and Discussion:
The methodological approach is not clear enough to fully understand the experimental design and results.
Regarding the characteristics of the rice flour—composition, degree of starch damage, and particle size—there is no need to discuss these results since the aim of the study was to investigate the effect of polysaccharide thickeners on dough and bread properties. This result, together with the used methodology, should be moved to the Materials section.
Reply:
Thank you for your suggestion. Rice flour properties have been moved to the Materials section (Body P.2 L.12–17).
7.Why were the water absorption and gelation characteristics determined only on the rice flour and not on the rice flour with added polysaccharide thickeners? What would be the relevance of these results to the aim of the study?
Reply:
You have raised an important point; however, we believe that would be outside the scope of our paper because in the sample preparation of this study, various thickened polysaccharides were dissolved in distilled water and added to rice flour. Therefore, the characteristics of the raw material rice flour and the thickened polysaccharide solutions are reported separately. We consider this discussion meaningful because it is clear that these properties have an impact on the properties and shelf life of the dough and bread.
However, the preparation of the thickened polysaccharides in combination with rice flour without dissolving them in water is a separate issue to be considered in the future.
8.2.2.2. Bread: From the present method description, the procedure of dough preparation with added polysaccharide thickener solutions is not visible.
Reply:
Thank you for your suggestion. I have corrected “Polysaccharide thickener solutions with concentrations ranging from 0 to 1.0% were mixed with rice flour. The amount of water added ranged from 92.5 g (0.25% polysaccharide added) to 100 g (1.0% polysaccharide added), considering the water absorption rate of the polysaccharide thickening agent.“ to “The polysaccharide thickener solutions were mixed with the rice flour at concentrations ranging from 0.25% to 1.0%. The amount of distilled water added ranged from 92.5 g (0.25% polysaccharide) to 100 g (1.0% polysaccharide), considering the water absorption rate of the polysaccharide thickening agent.“ in the text(Body P.2 L.32–35). And further clarified in “For the rice-flour bread batter, 100 g of rice flour and each polysaccharide thickener solution (or 90 g distilled water for the control) were mixed using a mixer (Aikoh Kenmix Aikoh Premier KMM770, Aikosha Mfg. Co., Ltd.) at 35℃ and 150 rpm for 1 min.” (Body P.2 L.35–38)
9.2.5.5. Sensory evaluation: The description should include all aspects of sensory analysis (trained or semi-trained panel, sample handling, conditions, etc.).
Reply:
Thank you for your suggestion; the sensory evaluation method section was updated to include the required details and now reads “The samples were obtained after the bread was baked, cooled at 25℃, placed in an airtight polypropylene container and stored at 4℃ for 1 hours, and the sample were a muffin-shaped bread (7 cm in diameter at the top and 5 cm in diameter at the bottom) radially divided into 6 equal pieces, with each piece consisting of 3 slices. The three types of rice-flour bread with each polysaccharide thickening agent (1.0% additive of MTG, TG, or XT) were evaluated on a 7-point scale by a panel of 30 untrained female university students, with a control (no additive) as the standard.” (Body P.5 L.5–L.11)
10.There is a lack of statistical information in the figures/tables to indicate whether the variations are significant or not. There is also no information on the number of analyses performed to obtain an analytical result.
The discussion is poorly written; it must be improved and particularly placed in the context of previously published papers.
All in all, the quality of the present manuscript is not satisfactory.
Reply:
Thank you for your helpful evaluation of our paper; the number of repeats is now given in the figures where statistical significance is relevant, and letters are used to show where results are significant or not.(Figure 9-14). We have also inserted p<0.05 in the body of the text for those differences that were significant.
The discussion has been improved to include discussion on all results relating to their potential or actual effect on the bread-making characteristics and preservation properties, as well as sensory aspects, where applicable. Previous studies have been further discussed to put the results into context, and your suggestions have greatly helped to improve the quality of our paper.
Thank you for all your suggestions and we look forward to presenting an improved manuscript.

Reviewer 2 Report
Comments and Suggestions for Authors
Introduction:
What is the reason for the decrease in rice consumption in Japan?
2.1. materials
Please standardize the spelling of the polysaccharides used. Currently, there are lowercase or uppercase letters.
2.2.1. Polysaccharide thickener solution
What does the term…. "adding a small amount"……
2.2.2. Bread.
Page 3. How were the portions weighed?
"It was then degassed and poured into a muffin mold (silicon, 5 cm in diameter at the bottom and 7 cm in diameter at the top) in 35 g portions".
2.3.1. Compositions.
Shouldn't a literature reference be added?
"The protein content, starch content, and amylose/amylopectin ratio of rice flour were analyzed by the Japan Food Research Center" [x].
2.3.2. Degree of star damage
"The absorbance was measured at 510 nm to calculate the degree of starch damage". Please, give the name of the device on which the absorbance was measured.
2.5.4. Bread chromaticity
How many locations were marked?
2.6. Statistical analysis
Please describe this point in detail (e.g., What was compared, what was the level of significance).
2.7. Ethical Considerations
Enter the number/symbol. "This study was carried out with the approval of the "Japan Women's University Ethical Review Committee for Experimental Research on Human Subjects."
3. Results and Discussion
The authors presented the research results in the indicated chapter, but there is practically no discussion.
Please reword the Conclusions. The authors' research aimed not to investigate allergies related to the use of wheat flour and other allergies.
Author Response
添付ファイルを確認してください。
-------------------------
Dear Editors and Reviewers
Thank you very much for reviewing our manuscript and offering your valuable advice.
We have addressed your comments with point-by-point responses, and revised the manuscript accordingly.
Responses to the Comments by Reviewer 2:
1.Introduction:
What is the reason for the decrease in rice consumption in Japan?
Reply:
Thank you. We have added “Reasons for the decline in rice consumption include the change from a Japanese-style diet to a Western-style diet, the increasing applicability of bread and noodles as staple foods, and their increasing demand.” (Body P.1 L.8–10)
2.1. materials
Please standardize the spelling of the polysaccharides used. Currently, there are lowercase or uppercase letters.
Reply:
Thank you for pointing out this inconsistency. Suitable revisions have been made.(Body P.2 L.20)
2.2.1. Polysaccharide thickener solution
What does the term…. "adding a small amount"……
Reply:
This means that the correct amount is used as per the concentrations mentioned, e.g., at 1.0% polysaccharide thickener solution we have added 10 g of powder to 990 g of distilled water. We have changed “adding a small amount” to “adding a small quantity”. (Body P.2 L.28)
2.2.2. Bread.
Page 3. How were the portions weighed?
"It was then degassed and poured into a muffin mold (silicon, 5 cm in diameter at the bottom and 7 cm in diameter at the top) in 35 g portions".
Reply:
Thank you for your clarifying question. It was weighed using an electronic balance; this has been added into the manuscript.(Body P.3 L.7)
2.3.1. Compositions.
Shouldn't a literature reference be added?
"The protein content, starch content, and amylose/amylopectin ratio of rice flour were analyzed by the Japan Food Research Center" [x].
Reply:
Thank you for the question; the analyses were done by an external laboratory on the rice flour used, so there is not a literature reference for this. Additionally, ingredients have been removed as they have been moved to the materials section, as pointed out by another reviewer. (Body P.2 L.12-13)
2.3.2. Degree of starch damage
"The absorbance was measured at 510 nm to calculate the degree of starch damage". Please, give the name of the device on which the absorbance was measured.
Reply:
Thank you for picking that up, but as this was not within the aim of our investigation, ingredients analysis has been moved to the materials section, as suggested by another reviewer. These were done by an external lab, so we no longer quote the method of analysis in the manuscript. (Body P.2 L.14)
2.5.4. Bread chromaticity
How many locations were marked?
Reply:
For one sample, three points were measured in the center of the sample. We have adapted the section to:
“2.5.4. Bread color(Body P.4 L34)
The color was measured using the CIE L*a*b* color space [34] and a color spectrophotometer (ND-1001DP, Nippon Denshoku Kogyo Co., Ltd, Japan). The L* (lightness), a* (redness), and b* (yellowness) values were determined to evaluate the color of the bread surface, by measuring three points in the center of the sample (n = 3–7).” (Body P.4 L.35–38)
2.6. Statistical analysis
Please describe this point in detail (e.g., What was compared, what was the level of significance).
Reply:
Thank you for your suggestion to improve this section; it has been adapted to “Statistical processing of the measured objective characteristics of the samples and sensory evaluation results was conducted using the statistical analysis software Excel Statistics 2012. Further, one-way analysis of variance was performed, and the samples that exhibited significant differences were further tested by multiple comparisons using Scheffe’s method. The significance level for all tests was set to 5%.” (Body P.5 L.30-34)
2.7. Ethical Considerations
Enter the number/symbol. "This study was carried out with the approval of the "Japan Women's University Ethical Review Committee for Experimental Research on Human Subjects."
Reply:
Thank you for pointing this out. We have included the number.(Body P.6 L.2)
- Results and Discussion
The authors presented the research results in the indicated chapter, but there is practically no discussion.
Please reword the Conclusions. The authors' research aimed not to investigate allergies related to the use of wheat flour and other allergies.
Reply:
Thank you for the suggestion; the discussions have all been improved to include discussion on all results relating to their potential or actual effect on the bread-making characteristics and preservation properties, as well as sensory aspects, where applicable. We have adapted the conclusions as below:
“Gluten-free rice-flour bread hardens over time and possesses poor storage qualities, which are attributed to its short shelf life of only 3 d. In this study, the type and concentration of the added polysaccharide thickener were found to exert different effects on the bread-making properties and shelf life. Results indicated that the addition of 0.5% MTG/TG or 1% XT effectively prevented hardening. Further, the addition of TG or MTG to gluten-free rice-flour bread was more effective in bread making and preservation at lower concentrations than the addition of XT. MTG addition reduced the specific volume of the bread but increased the moisture content and retention. It also reduced the stress at 70% compression and was found to be soft, moist, and preferred over the other additives and control, demonstrating the positive impact of the thermoresponsive xyloglucan additive. Further investigations into the bread-making properties of gluten-free rice-flour bread are essential for improving preferable qualities and preservation characteristics, as they could greatly help people suffering from wheat-flour and other allergies, as well as celiac disease, by providing good alternative breads and baked goods based on rice flour.” (Body P.19 L.14-26)

Reviewer 3 Report
Comments and Suggestions for Authors
This work is very interesting, not only from the scientific-technological point of view but also for consumers, especially for celiacs, as well as having a clear practical application. Thank you very much for being able to share and be able to evaluate this work However, there is some point that should be reviewed in depth since it is poorly described from the methodological point of view. I recommend authors use the original CIELAB color system source and some of the guidelines color evaluation that exists for various types of foods.
2.5. Bread chromaticity "The L* (brightness), a* (redness), and b* (yellowness) values were determined by using a colorimeter (ND-1001DP, Nippon Denshoku Kogyo Co., Ltd.) [35] to evaluate the color of the bread surface". On this point, the Authors must be more precise. In my opinion, the use of chromaticity is not correct, could be Bread color, chromaticity is a color property. L* is lightness. At this point other characteristics must be mentioned, such as illuminant, standard observer, sample thickness, and the infinite solid was determined. Which type of background was used? How samples were measured? optically pure glass was used? and how do the authors use it?
In statistical methods, authors must specify factors and levels. Also the significance level.
3.4.3. Bread Color. Authors only make a description. But, there are not any comments about their results. Discussion about their results is absolutely necessary since during bread making process several changes take place and these changes directly affect color properties.
Author Response
原稿をレビューし、貴重なアドバイスをいただき、誠にありがとうございます。
私たちはあなたのコメントにポイントごとの応答で対処し、それに応じて原稿を修正しました。

Round 2
Reviewer 1 Report
Comments and Suggestions for Authors
The paper can be accepted in the present form
Author Response
Thank you.
Reviewer 2 Report
Comments and Suggestions for Authors
Thank the authors for considering the reviewer's comments and the changes made. I suggest adding, where necessary, literature references to the introduced fragments, e.g., "Reasons for the decline in rice consumption include the change from a Japanese-style diet to a Western-style diet, the increasing applicability of bread and noodles as staple foods, and their increasing demand" (Introduction).
Author Response
原稿をレビューし、貴重なアドバイスをいただき、誠にありがとうございます。
私たちはあなたのコメントにポイントごとの応答で対処し、それに応じて原稿を修正しました。
----------------------------------
Dear Editors and Reviewers
Thank you very much for reviewing our manuscript and offering your valuable advice.
We have addressed your comments with point-by-point responses, and revised the manuscript accordingly.
Responses to the Comments by Reviewer 2:
1.Introduction:
I suggest adding, where necessary, literature references to the introduced fragments, e.g., "Reasons for the decline in rice consumption include the change from a Japanese-style diet to a Western-style diet, the increasing applicability of bread and noodles as staple foods, and their increasing demand" (Introduction).
Reply:
Thank you for your suggestion . We added references [4],[5], and [6] as the most recent data and changed from “Reasons for the decline in rice consumption include the change from a Japanese-style diet to a Western-style diet, the increasing applicability of bread and noodles as staple foods, and their increasing demand.” to “Reasons for the decline in rice consumption include the change from a Japanese-style diet of rice and fish to a Western-style diet with increased intake of meat [4] and fat [5], the increasing applicability of bread and noodles [6] as staple foods, and their increasing demand.” (Body P.1 L.8–10)
